# Adhesion and Proliferation of Mesenchymal Stem Cells on Plasma-Coated Biodegradable Nanofibers

Anton M. Manakhov [1,*], Anastasiya O. Solovieva [1], Elizaveta S. Permyakova [1,2], Natalya A. Sitnikova [1], Lyubov S. Klyushova [3], Philipp V. Kiryukhantsev-Korneev [2], Anton S. Konopatsky [2] and Dmitry V. Shtansky [2]

1   Laboratory of Pharmacological Active Compounds, Research Institute of Clinical and Experimental Lymphology—Branch of the ICG SB RAS, 2 Timakova Str., 630060 Novosibirsk, Russia; solovey_ao@mail.ru (A.O.S.); permyakova.elizaveta@gmail.com (E.S.P.); sitnikovanat9@gmail.com (N.A.S.)

2   Laboratory of Inorganic Nanomaterials, National University of Science and Technology "MISiS", Leninsky Prospekt 4, 119049 Moscow, Russia; kiruhancev-korneev@yandex.ru (P.V.K.-K.); ankonopatsky@gmail.com (A.S.K.); shtansky@shs.misis.ru (D.V.S.)

3   Institute of Molecular Biology and Biophysics, Federal Research Center of Fundamental and Translational Medicine (IMBB FRC FTM) 2/12, Timakova Str., 630117 Novosibirsk, Russia; klyushovals@mail.ru

*   Correspondence: ant-manahov@ya.ru

**Abstract:** Various biomedical applications of biodegradable nanofibers are a hot topic, as evidenced by the ever-increasing number of publications in this field. However, as-prepared nanofibers suffer from poor cell adhesion, so their surface is often modified. In this work, active polymeric surface layers with different densities of COOH groups from 5.1 to 14.4% were successfully prepared by $Ar/CO_2/C_2H_4$ plasma polymerization. It has been shown that adhesion and proliferation of mesenchymal stem cells (MSCs) seeded onto plasma-modified PCL nanofibers are controlled by the $CO_2:C_2H_4$ ratio. At a high $CO_2:C_2H_4$ ratio, a well-defined network of actin microfilaments is observed in the MSCs. Nanofibers produced at a low $CO_2:C_2H_4$ ratio showed poor cell adhesion and very poor survival. There were significantly fewer cells on the surface, they had a small spreading area, a poorly developed network of actin filaments, and there were almost no stress fibrils. The maximum percentage of proliferating cells was recorded at a $CO_2:C_2H_4$ ratio of 35:15 compared with gaseous environments of 25:20 and 20:25 ($24.1 \pm 1.5$; $8.4 \pm 0.9$, and $4.1 \pm 0.4\%$, respectively). Interestingly, no differences were observed between the number of cells on the untreated surface and the plasma-polymerized surface at $CO_2:C_2H_4 = 20:25$ ($4.9 \pm 0.6$ and $4.1 \pm 0.4$, respectively). Thus, $Ar/CO_2/C_2H_4$ plasma polymerization can be an excellent tool for regulating the viability of MSCs by simply adjusting the $CO_2:C_2H_4$ ratio.

**Keywords:** MSC; plasma; nanofibers; cell viability; COOH groups; XPS; surface; biodegradable nanocomposites

## 1. Introduction

In the field of tissue engineering, in recent years there has been a growing interest in porous structures consisting of biodegradable fibers, the structure of which is similar to the extracellular matrix (ECM). This is confirmed by a significant number of publications. As shown in Figure 1a, the number of publications containing the keywords "Cell", and "Adhesion" and "Nanofibers" exceeded 350 per year, although until the 2000s this topic was not covered at all. Indeed, recent technological progress enabled the production of nanofibrous mats on a pilot scale at a reasonable cost using innovative methods, including electrodeless electrospinning. As a result, electrospun nanofibers have become an attractive and affordable technical solution for tissue engineering.

An analysis of the frequency of use of various keywords when searching in the Scopus database for the keywords "CELLS & ADHESION & NANOFIBERS" clearly shows the most popular directions in the modern scientific literature in the field of biodegradable

nanofibers (Figure 2a). The most relevant keywords are Electrospinning (a method for producing electrospun nanofibers), Scaffold, Tissue Engineering, and Polycaprolactone. Indeed, the use of nanofibers for tissue engineering has become a topical and very attractive field due to the high potential of these materials. The use of polylactic acid (PLA) and polycaprolactone (PCL) have gained significant interest during the last years, and various biomedical and smart materials with outstanding properties have been reported [1,2]. Indeed, PCL possesses good mechanical properties and long-term stability from a few months up to 3 years in vivo and it is approved by U.S. Food and Drug Administration. Thus, recently PCL has become the material of choice for biomedical materials.

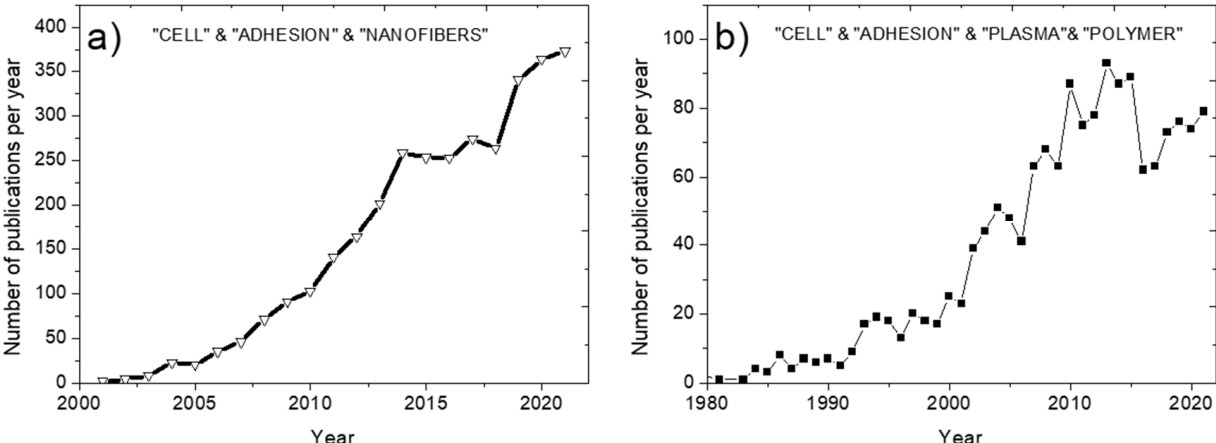

**Figure 1.** Increase in the number of publications by years. Search in Scopus by sections Title, Abstract, Keywords: "CELL & ADHESION & NANOFIBERS" (**a**) and "PLASMA & POLYMER & CELL & ADHESION" (**b**).

Figure 2a highlights the main areas of research that have made a significant contribution to this field. The most promising method for obtaining biodegradable nanofibers is electrospinning from polymer solutions [3,4]. Electrospinning of nanofibers is possible from a solution of both natural polymers (collagen, gelatin, chitosan) and synthetic ones (PCL, polyethylene glycol, etc.). Nanofibers obtained from natural polymers demonstrate high biocompatibility; however, obtaining stable and homogeneous nanofibers is a rather tricky task [5]. For a long time, it was almost impossible to obtain pure collagen and chitosan nanofibers and, therefore, fibers consisting of a mixture of polymers (for example, chitosan/polyethylene oxide) were obtained [3,4]. In addition, collagen is expensive, and collagen and gelatin nanofibers are often unstable in an aquatic environment. Therefore, the development of bioactive nanofibers of synthetic polymers is an auspicious task. However, most of these polymeric nanofibers are superhydrophobic. Therefore, they must be additionally processed to enhance adhesion and cell proliferation on their surfaces [6,7].

Another frequently encountered keyword is "Mesenchymal stem cells" (MSC). This indicates a high interest in in vitro studies of MSC adhesion to the nanofibers. Indeed, this type of cell is often used as a model system for testing the biocompatibility of various materials [8].

To date, the most common methods for modifying nanofibers are (1) liquid treatment of nanofibers (for example, soaking in a KOH solution), (2) co-spinning of biopolymers (gelatin or collagen), (3) plasma treatment in a gas discharge in combination with (or without) grafting growth factors, (4) plasma polymerization [9,10]. The first method is the least promising, since liquid processing leads to the degradation of nanofibers [11]. The disadvantages of the second method are its non-universality and too limited possibilities, since the surface of the nanofibers does not contain active groups to which active substances (for example, growth factors or antibiotics) could be additionally attached. The third method, plasma treatment (treatment in a gas discharge, for example, in air, oxygen,

or argon), is an environmentally friendly and energy efficient method. Attachment of growth factors to plasma-treated PCL and PLA nanofibers significantly accelerated wound healing [12].

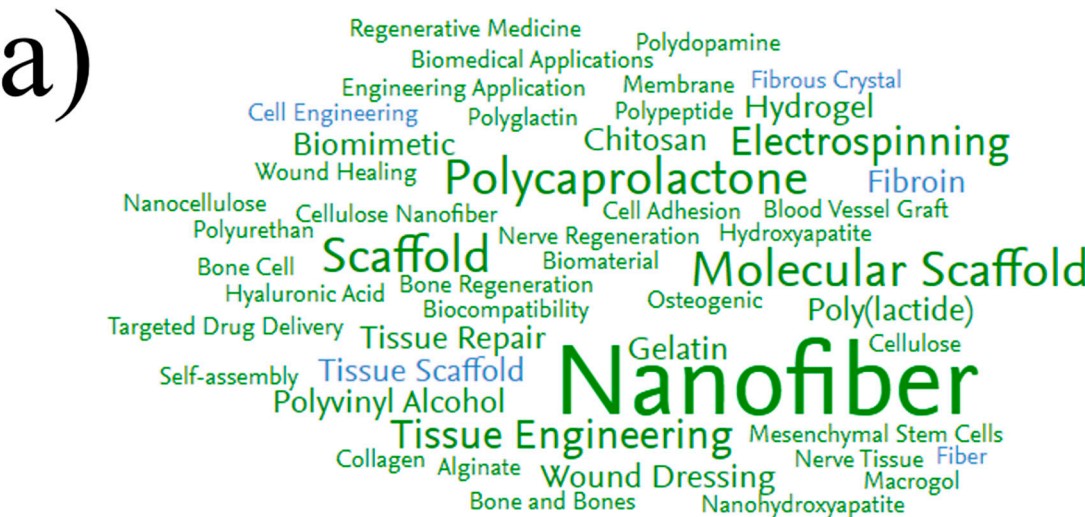

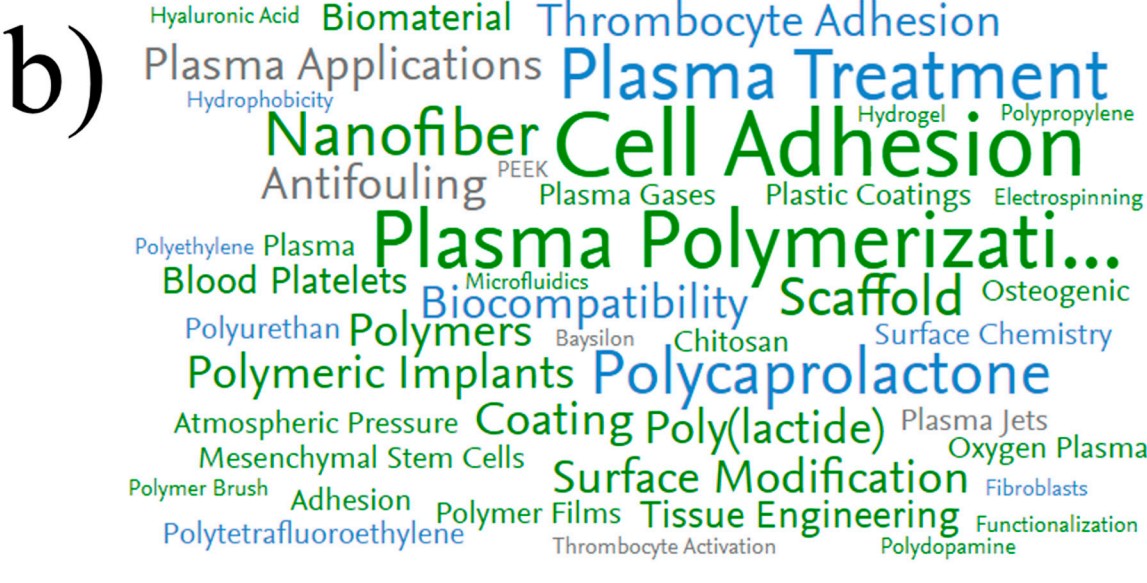

**Figure 2.** Keyword distributions by frequency of use when searching in Scopus for the keywords "CELL & ADHESION & NANOFIBERS" (**a**) and "PLASMA & POLYMER & CELL & ADHESION" (**b**). Font size corresponds to keyword relevance (larger font size corresponds to higher frequency of the keyword appearance in the dataset). The color corresponds to the decreasing (blue) or increasing (green), or does not changing (grey) current trend of the keyword usage.

The use of plasma polymers to stimulate cell adhesion started in the 1980s (Figure 1b), but their application for the modification of nanofibers is a more recent topic (Figure 2b). Today, this field is developing rapidly (big green keyword "Nanofiber"). Note that plasma treatment and plasma polymerization have significant differences. Since the effect of plasma treatment decreases very quickly (within 1–2 days), various growth factors are often grafted onto the material surface to improve bioactivity. However, it is not yet known

exactly how long the activity of immobilized growth factors can be maintained. Plasma polymerization, i.e., the deposition of plasma polymers under the action of a discharge in vapors of organic monomers, is an alternative promising approach. The deposited layers of plasma polymers are highly stable [13,14]. In addition, the concentration of active surface groups in plasma polymers is always higher than in plasma-treated surfaces.

The blue color of the keyword "Plasma treatment" (Figure 2b) indicates that the number of publications in this field is decreasing every year, and the green keyword "Plasma polymer" indicates an increase in the number of publications in this field.

The combination of electrospinning and plasma polymerization technologies is an excellent technological approach to the synthesis of promising materials for tissue engineering and wound dressings. Figure 2b shows a variety of polymeric substrates (polylactide, polycaprolactone, polyurethane, etc.), as well as cell cultures that can be cultivated on plasma-modified polymers. As for the type of functional groups deposited by the plasma, it includes carboxyls [15,16], amines [13,17], epoxides [18,19], hydroxyls [20], thiols [21], aldehydes [22], and others [23].

The range of applications of COOH-modified PCL nanofibers is quite extensive. For example, polycaprolactone nanofibers can be used to replace bone tissue. However, this requires preliminary mineralization of the nanofibers (for example, due to the growth of a hydroxyapatite film) by immersing nanofibers in a synthetic physiological solution (SPS) which contains calcium and sodium salts. The growth of hydroxyapatites can be accelerated if the nanofibers are modified with COOH groups [1,4,24,25]. The obtained nanocomposites can be attractive candidates for bone implants or bone replacement/regeneration materials. In addition, PCL-coated COOH nanofibers are promising candidates for local drug delivery to treat bone infection. For example, gentamicin-loaded nanofibers improved the healing of bone fractures and prevented tissue inflammation [26–28].

The deposition of COOH coatings by plasma polymerization can be carried out in various ways: using low or atmospheric pressure plasma polymerization in various gas mixtures. Acrylic acid is the most common precursor used for the deposition of COOH coatings [29–32]. However, acrylic acid plasma polymers are poorly stable in water [33]. More promising results were obtained using plasma polymerization of maleic anhydride in a mixture with acetylene or vinyltrimethoxysilane [16,34–36]. These films had a high density of carboxyl groups and low water solubility (less than 5% after 24 h in water or phosphate buffer). Plasma polymerization in an argon/ethylene/$CO_2$ mixture is also a promising method, since such films are highly stable [37,38]. However, the mechanism of plasma polymerization and the concentration of COOH groups in the films have not been fully studied. To achieve high stability and uniform coating of nanofibers with a polymer layer, improvement of the methodology is required.

This work was designed in such a way as to gain new knowledge and highly demanded results on the development of nanofibrous biodegradable composites consisting of PCL nanofibers and $Ar/CO_2/C_2H_4$ plasma polymers. After careful analyzing of emerging trends, we selected the most promising approaches, chose the most commonly used cell type, and utilized an environment-friendly surface modification technology. We then carefully investigated the influence of plasma gas mixtures on the chemical compositions of plasma layers and the influence of surface composition on cell behavior.

## 2. Materials and Methods

### 2.1. Electrospinning PCL Nanofibers

Nanofibers were prepared by electrospinning a 9 wt% solution of PCL (80,000 g/mol). Sample processing can be found elsewhere [39]. Briefly, the granulated PCL was dissolved in acetic acid (99%) and formic acid (98%). All compounds were purchased from Sigma Aldrich (Darmstadt, Germany). The weight ratio of acetic acid (AA) to formic acid (FA) was 2:1. The PCL solutions in AA and FA were stirred at 25 °C for 24 h and then subjected to electrospinning with 20 cm long wired electrodes using a Nanospider™ NSLAB 500 machine (ELMARCO, Liberec, Czech Republic). The applied voltage was 50 kV. The

distance between the electrodes was set to 100 mm. The as-prepared and non-treated PCL nanofibers are referred to in the text as PCL-ref.

### 2.2. Plasma COOH Coating

The $Ar/CO_2/C_2H_4$ plasma polymerization methodology is described in detail elsewhere [37,40]. Briefly, the COOH plasma polymer layers were deposited onto Si wafers and PCL nanofibers using a UVN-2M vacuum system equipped with rotary and oil diffusion pumps. The residual pressure in the reactor was below $10^{-3}$ Pa. The plasma was ignited using a radio frequency (RF) Cito 1310-ACNA-N37A-FF power supply (Comet, Flamatt, Switzerland) connected to an RFPG-128 disk generator (Beams and Plasmas, Russia) installed in the vacuum chamber. Duty cycle and RF power were set to 5% and 500 W, respectively. $CO_2$ (99.995%), Ar (99.998%), and $C_2H_4$ (99.95%) were fed into the vacuum chamber. The gas flow of Ar was set to 50 sccm, while the flows of $CO_2$ and $C_2H_4$ varied in the ranges from 20 to 40 and from 5 to 25 sccm, respectively. They were controlled using a 647C Multi-Gas Controller (MKST, Newport, RI, USA). The chamber pressure was measured with a VMB-14 unit (Tokamak Company, Dubna, Russia) and D395-90-000 BOC Edwards controllers. The distance between the RF-electrode and the substrate was set to 8 cm. The deposition time was set to 15 min. The samples with $CO_2$:$C_2H_4$ ratios of 40:5, 35:10, 35:15, 25:20, and 20:25 were prepared and investigated. They are denoted according to the $CO_2$:$C_2H_4$ flows ratio.

### 2.3. Characterization of Samples

The samples morphology was examined by scanning electron microscopy (SEM). SEM analysis was carried out with a JSMF 7600 microscope (JEOL Ltd., Tokyo, Japan) equipped with an energy-dispersive X-ray spectrometer. To compensate for surface charge, the samples were coated with a ~5 nm thick Pt layer. The average NFs diameter was determined using the ImageJ software based on 100 measurements.

The sample chemical characterization was performed by X-ray photoelectron spectroscopy (XPS), energy-dispersive X-ray spectroscopy (EDXS), and Fourier-transformed infrared (FTIR) spectroscopy. The XPS analysis was carried out using a PHI5500VersaProbeII instrument (PHI) equipped with a monochromatic Al Kα X-ray source (hν = 1486.6 eV) at a pass energy of 23.5 eV and X-ray power of 50 W. The spectra were fitted using the CasaXPS software after subtracting the Shirley-type background. The maximum lateral resolution of analyzed area was 0.7 mm. The binding energies (BEs) for all carbon and oxygen environments were taken from the literature [26,41–43]. The BE scale was calibrated by setting the CHx component at 285 eV. FTIR spectra (100 scans) in the spectral range from 370 to 4000 $cm^{-1}$ were recorded in increments of 4 $cm^{-1}$ on a Vertex 80v FTIR spectrometer (Bruker, Billerica, MA, USA) with a parallel beam transmittance accessory. The spectra were collected at room temperature (20–25 °C). The maximum lateral dimension of the analyzed area was 0.7 mm.

The sample wettability was assessed by measuring the water contact angle (WCA). The measurements were carried out on an Easy Drop Kruss (KRÜSS, Hamburg, Germany) device. For each sample, at least five WCA measurements were performed.

### 2.4. Cell Tests

Human MSCs (4–6 passages) were taken from the culture bank of Research Institute of Clinical and Experimental Lymphology (RICEL), a branch of the Institute of Cytology and Genetics, Siberian Branch of the Russian Academy of Sciences, which were extracted from the bone marrow as described elsewhere (the study was approved by the Ethics Committee of the RICEL-branch of ICG SBRAS (No 115 from 24 December 2015) [44]. Cells were cultured in DMEM/F12 Medium (Sigma Aldrich, Paisley, UK) and supplemented with 10% fetal bovine serum (FBS, Gibco, Carlsbad, CA, USA) under standard culture conditions (humidified atmosphere, 5% $CO_2$ and 95% air, at 37 °C). Cells were seeded on round-shaped (5 mm diameter) scaffolds at a concentration of $20 \times 10^3$ cells in a volume of 20 μL. Cell

adhesion after 2 h was assessed by cell area after phalloidin staining of cell actin filaments (short-time adhesion). Hoechst staining of cell nuclei was used to count cells and identify apoptotic cells by nuclear morphology. Cell proliferative activity was assessed using the Click-iT™ EdU Cell Proliferation Kit for Imaging (ThermoFisher Scientific, Eugene, OR, USA) according to the manufacturer's protocol recommendations.

An IN Cell Analyzer 2200 (GE Healthcare, Amersham, UK) was used to perform automatic imaging of six fields per well at 200 (for cell counting and determination of the average cells size, the entire surface of round-shaped (5 mm diameter) scaffolds was photographed, in 3 repeats) and 600 magnification (for better visualization of the shape of cells and the nature of the formation of active filaments) in fluorescence channels. The resulting images were used to analyze cell number and cell area using the IN Cell Investigator software (GE Healthcare, Amersham, UK).

## 3. Results

### 3.1. The Morphology and Wettability of $Ar/CO_2/C_2H_4$ Plasma Polymerized Layers

The SEM micrograph of sample PCL-ref (Figure 3a) revealed a homogenous structure of randomly oriented PCL nanofibers with an average diameter of $228 \pm 37$ nm. The deposition of $Ar/CO_2/C_2H_4$ plasma polymer layers led to only a slight increase in fiber diameters (Figure 3b–d). The mean diameter nanofibers coated by the $Ar/CO_2/C_2H_4$ layers deposited at $CO_2$:$C_2H_4$ ratios of 35:15, 25:20, and 20:25 was $294 \pm 98$, $249 \pm 48$, and $312 \pm 123$ nm, respectively. After deposition of the polymer layer, the morphology of the nanofibers and their smooth surface were preserved.

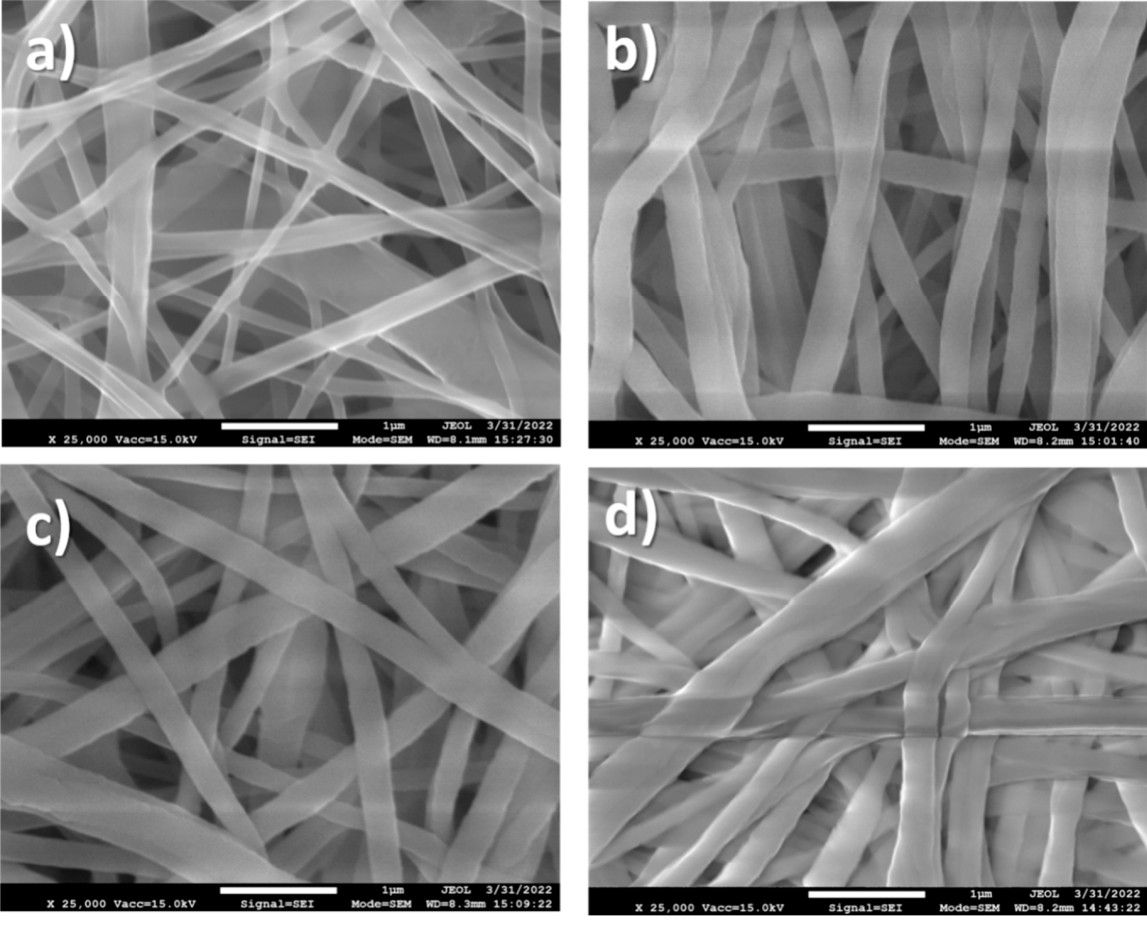

**Figure 3.** SEM micrographs of PCL-ref (**a**), $Ar/CO_2/C_2H_4$ plasma polymers deposited at $CO_2$:$C_2H_4$ of 20:25 (**b**), 25:20 (**c**), and 35:15 (**d**).

The PCL-ref showed a WCA of $117 \pm 1.3°$, which corresponds to a hydrophobic surface. The deposition of $Ar/CO_2/C_2H_4$ plasma polymerized layers led to significant changes in surface wettability. The deposition of plasma polymerized layers at low $CO_2$:$C_2H_4$ ratio led to a very slight decrease in WCA from $117 \pm 1.3°$ to $103.5 \pm 9.2°$ (Figure 4). An increase in $CO_2$:$C_2H_4$ from 0.8 to 1.25 allowed one to reduce the WCA to $44.0 \pm 4.6°$, and further increase in this ratio led to super hydrophilic surfaces with a WCA of $7.4°$. It is most likely that these large WCA variations are associated with the surface chemistry of the deposited layers.

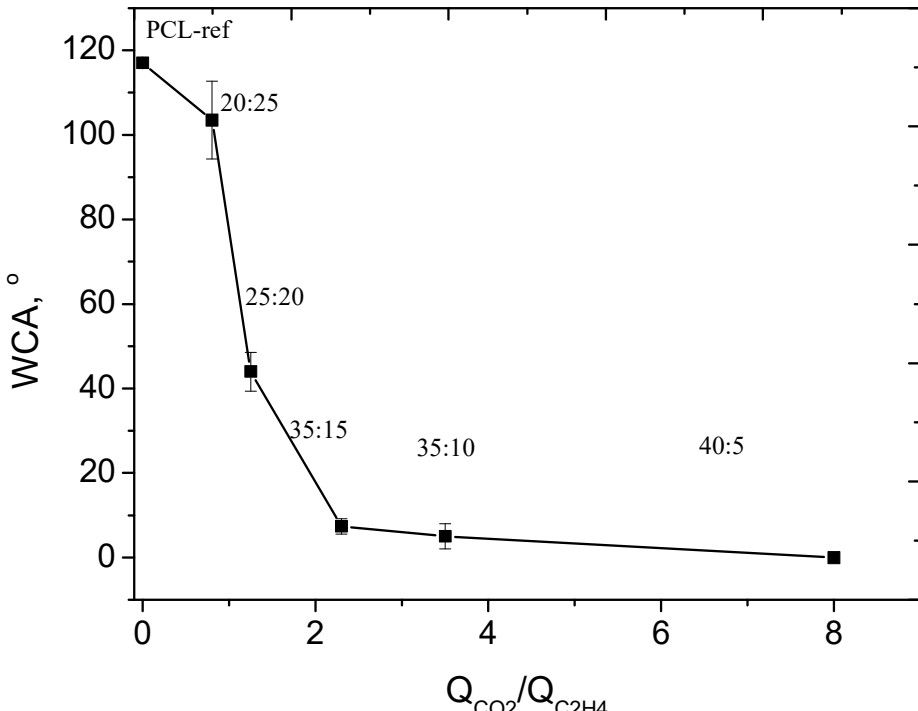

**Figure 4.** The dependence of WCA on $CO_2$:$C_2H_4$ ratio ($Q_{CO2}/Q_{C2H4}$).

### 3.2. The Control of Surface Chemistry by Adjustment of Precursor's Ratio

3.2.1. FT-IR Results

According to the FTIR spectra presented in Figure 5, the $Ar/CO_2/C_2H_4$ plasma polymerized layers exhibited the features of hydrocarbons and carboxylic acid/ester groups. The peak at $1730\ cm^{-1}$ is assigned to C=O stretching of the carboxylic acid or ester groups. The peaks located at 2885, 2935, and $2975\ cm^{-1}$ are attributed to C-$H_2$ asymmetric stretching, C-$H_3$ asymmetric stretching, and $CH_2/CH_3$ symmetric stretching, respectively. The peaks at 1450 and $1380\ cm^{-1}$ are attributed to H-C-H bending vibrations. Although all the above-mentioned features are clearly visible for all FT-IR spectra regardless of plasma conditions, some trends related to the $CO_2$:$C_2H_4$ ratio are also evident. The intensity of the $CH_2$ and $CH_3$ peaks decreases with an increase in the $CO_2$:$C_2H_4$ ratio, while the C=O peak gains its intensity. To demonstrate the effect of $CO_2$:$C_2H_4$ ratio on different incorporation of C(O)O and C-H groups, we plotted the ratio of the integrated intensities (area under the peaks) of the C-H ($I_{CH}$) and C=O ($I_{C=O}$) peaks as a function of the $CO_2$:$C_2H_4$ ratio (Figure 6). The results for layer deposited at 40:5 ratio was not used due to very noisy spectra. There is a clear trend towards an increase in $I_{C=O}/I_{CH}$, with increasing $Q_{CO2}/Q_{C2H4}$ over a wide range of $CO_2$:$C_2H_4$ ratios. This trend was further confirmed by XPS analysis.

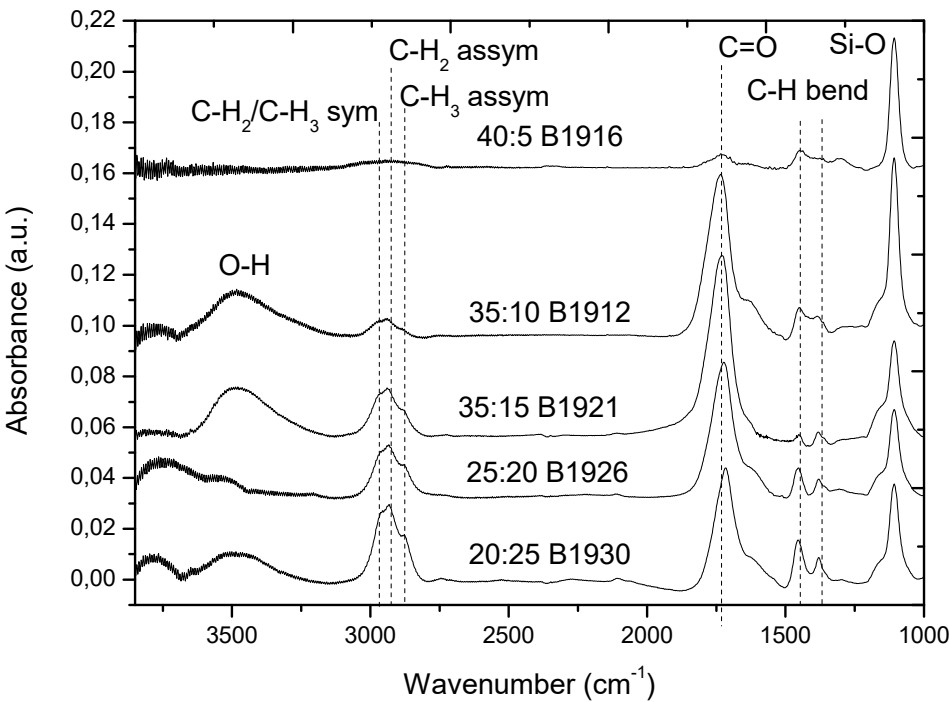

**Figure 5.** FTIR spectra of $Ar/CO_2/C_2H_4$ layers deposited at different $CO_2$:$C_2H_4$ ratios onto Si wafers.

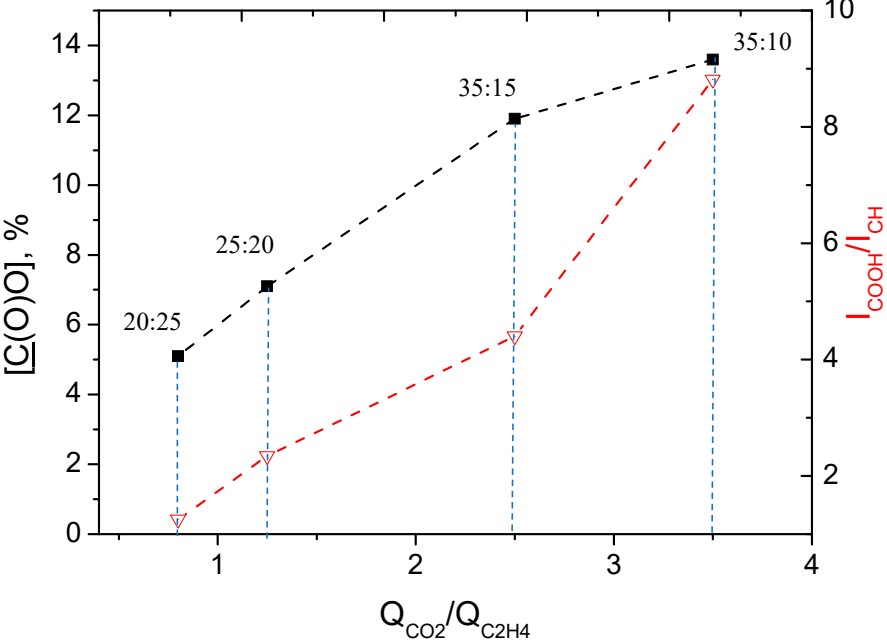

**Figure 6.** Dependence of C=O/CH peak intensity ratio ($I_{C=O}/I_{CH}$) and C(O)O concentration on $Q_{CO2}/Q_{C2H4}$.

### 3.2.2. XPS Results

XPS analyses performed on both Si wafers and PCL nanofibers revealed that all samples were composed of only carbon and oxygen. As shown in Table 1, carbon and oxygen concentrations correlate with the $CO_2$:$C_2H_4$ ratio, and the oxygen percentage increases with $Q_{CO2}/Q_{C2H4}$. This is not surprising, since an increase in plasma $CO_2$ concentration increases the content of oxygenated species. To understand how the carbon environment correlates with the $CO_2$:$C_2H_4$ ratio, the XPS C1s curve was fitted.

**Table 1.** Composition of samples (in at. %) derived from the XPS analysis.

| $CO_2$:$C_2H_4$Ratio | O | C |
|---|---|---|
| 25:20 (Si wafer) | 9.7 | 90.3 |
| 20:25 (Si wafer) | 22.5 | 77.5 |
| 35:15 (Si wafer) | 27.7 | 72.3 |
| 35:10 (Si wafer) | 30.8 | 69.2 |
| 20:25 (PCL) | 24.6 | 75.4 |
| 35:15 (PCL) | 27.5 | 72.5 |
| PCL-ref | 26.1 | 73.9 |

All samples were deconvoluted using five components: hydrocarbons $CH_x$ (BE = 285.0 eV, used for BE scale calibration, FWHM = 1.4 ± 0.1), carbon neighbored to carboxylic acid or ester group $C^*$-C(O)O (BE = 285.5 ± 0.05 eV, FWHM = 1.3 ± 0.15), carbon singly bonded to oxygen C-O (BE = 286.55 ± 0.05 eV, FWHM = 1.4 ± 0.15), carbon doubly bonded to oxygen C=O/O-C-O (BE = 287.9 ± 0.1 eV, FWHM = 1.4 ± 0.1), and ester carbon or carboxylic group C(O)O (BE = 289.0 ± 0.05 eV, FWHM = 1.35 ± 0.05). The curve fitting and concentrations of all components are shown in Figure 7.

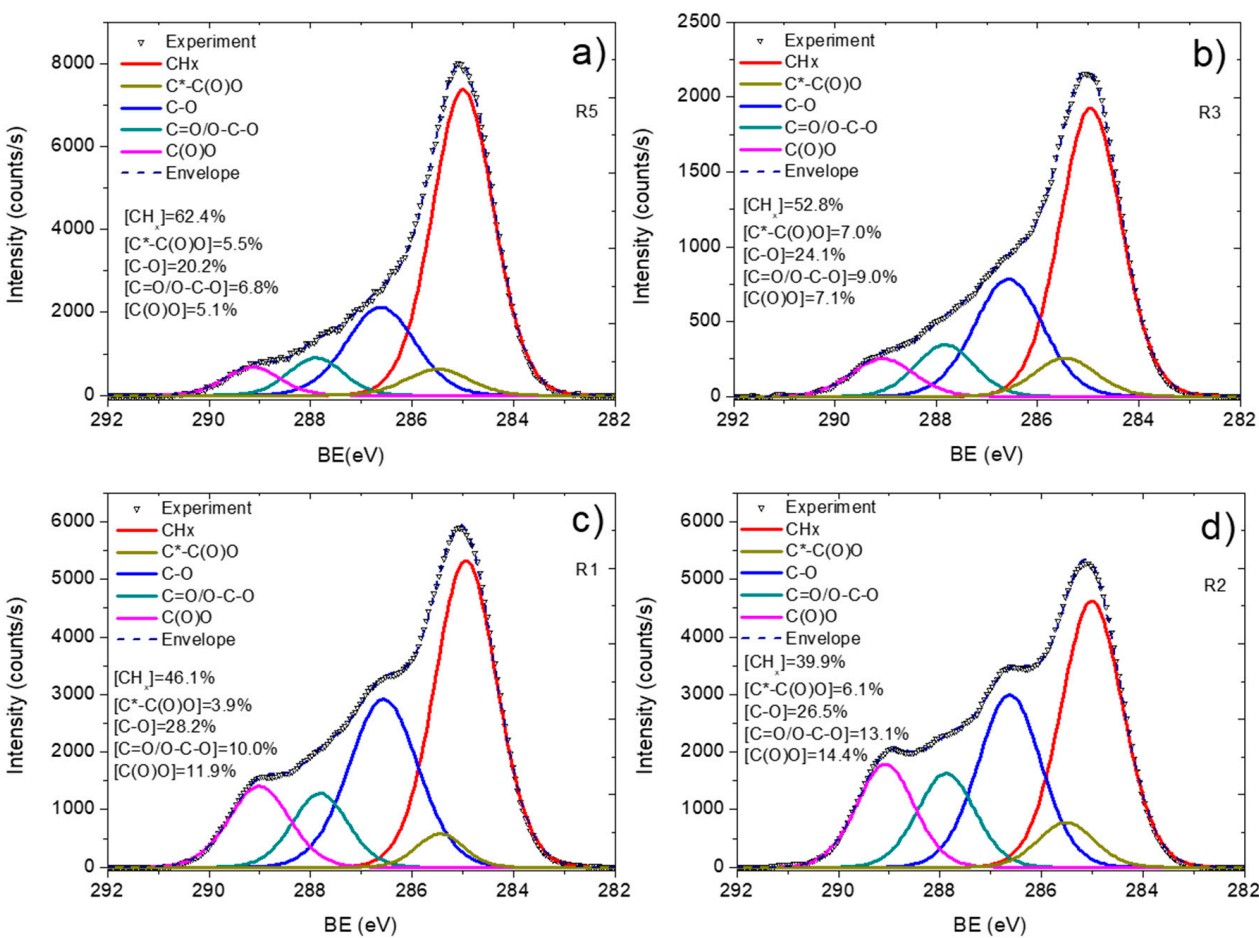

**Figure 7.** C1s XPS curve fitting of Ar/$CO_2$/$C_2H_4$ plasma polymerized layers deposited onto Si wafer at different $CO_2$:$C_2H_4$ ratios: 20:25 (**a**), 25:20 (**b**), 35:15 (**c**), and 35:10 (**d**).

The layer deposited at lowest $CO_2$:$C_2H_4$ ratio (20:25) shows a C1s spectrum with a dominating $CH_x$ contribution (62.4%) and a very low C(O)O concentration. An increase in the $CO_2$:$C_2H_4$ ratio from 20:25 to 25:20 led to a significant decrease in the $CH_x$ component, and it further decreased for the layers deposited at 35:15 (46.1%) and 35:10 (39.9%). The C(O)O component increased almost 3 times, changing the ratio from 20:25 to 35:10. The

dependence of C(O)O concentration on $CO_2$:$C_2H_4$ is plotted in Figure 6. It is similar to that observed for the $I_{C=O}/I_{CH}$ ratio as a function of $CO_2$:$C_2H_4$. Therefore, both FTIR and XPS revealed similar influence of the plasma gas composition on the layer chemistry. The observed difference in the surface chemistry also correlated with the WCA results. Indeed, the layer deposited at 20:25 exhibited a high WCA due to the predominance of hydrocarbon environment. An increase in the concentration of oxygenated species in the plasma allowed one to deposit a layer with higher concentrations of C(O)O and other oxygenated components, which led to a significant decrease in the water contact angle. Thus, by simply adjusting the $CO_2$:$C_2H_4$ ratio, we were able to synthesize layers with similar morphologies but very different surface chemistries and wettability.

It is worth noting that the elemental and functional compositions of the $Ar/CO_2/C_2H_4$ plasma polymer layers deposited onto Si wafers and PCL nanofibers were very similar (Table 1). To conform the successful surface functionalization of PCL nanofibers, the C1s XPS spectra of PCL-ref and $Ar/CO_2/C_2H_4$ plasma polymerized layer deposited with a $CO_2$:$C_2H_4$ ratio of 35:15 iscompared in Figure 8.

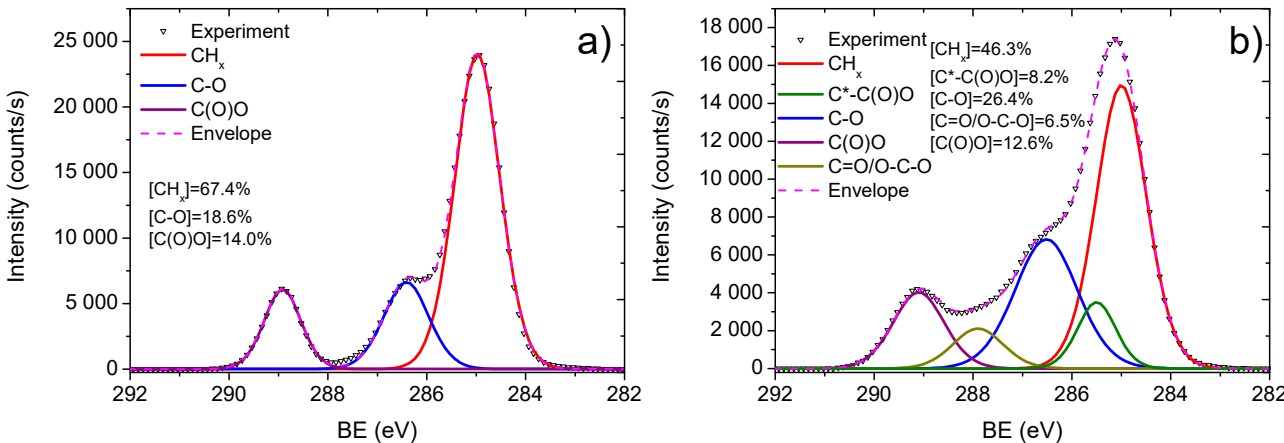

**Figure 8.** XPS C1s spectra of PCL-ref (**a**) and plasma polymerized $Ar/CO_2/C_2H_4$ layer deposited at $CO_2$:$C_2H_4$ ratio of 35:15 onto PCL nanofibers (**b**).

To confirm that a layer deposited onto PCL nanofibers would have the same functional composition as on Si and would uniformly coat the nanofibrous morphology, we performed XPS analyses on PCL-ref and PCL nanofibers coated with a plasma layer. The XPS C1s spectrum of PCL-ref was fitted with a sum of three components, namely hydrocarbons $CH_x$ (BE = 285 eV), ether group C-O (BE = 286.4 eV), and ester group C(O)O (BE = 289.0 eV) (Figure 5a). The full width at the half maximum (FWHM) of C-O was set to 1.35 eV, while the FWHMs of $CH_x$ and C(O)O components were 1.1 and 0.95 eV, respectively.

The C1s XPS spectrum of plasma polymerized layer obtained at $CO_2$:$C_2H_4$ = 35:15 was fitted with the sum of five components: hydrocarbons $CH_x$ (BE = 285.0 eV, used for BE scale calibration), carbon neighbored to carboxylic acid or ester group C*-C(O)O (BE = 285.5 $\pm$ 0.05 eV), carbon singly bonded to oxygen C-O (BE = 286.55 $\pm$ 0.05 eV), carbon doubly bonded to oxygen C=O/O-C-O (BE = 288.0 $\pm$ 0.05 eV), and carbon of ester or carboxylic group C(O)O (BE = 289.2 $\pm$ 0.03 eV). The concentrations of all components are reported in Figure 8.

### 3.3. Biological Tests of Plasma Polymers with Varying COOH Density

Surface modification is an effective tool for improving the interaction between cells and biomaterial. To assess the effect of surface modifications on the functional activity of cells, first of all, the adhesion of MSCs to the studied surfaces was determined. The actin cytoskeleton plays an important role in the regulation of cellular activity. The dynamic regulation of cytoskeletal synthesis, remodeling, and function is critical to many physiological processes and is integral to successful regeneration. In the presented work, the dynamics

of the actin cytoskeleton, the formation of lamellipodia and actin-rich filopodia, as well as actin-rich areas at the site of cell contact with the surface and the cells' spreading area, were evaluated.

The spreading area of the cells was estimated 2 h after their seeding on the nanofibers by Phalloidin staining of actin filaments. The results obtained indicate that the cell area was maximal on the PCL-ref sample (Figure 9). However, it was noted that there are many pieces of cell membranes on the untreated fiber (white arrows on the photo), which indicates weak cell adhesion, followed by their detachment from the surface. We suggest that these cells, trying to attach to the hydrophobic surface, grope for suitable sites, which leads to the formation of lamellipodia and actin-rich filopodia (Figure 10).

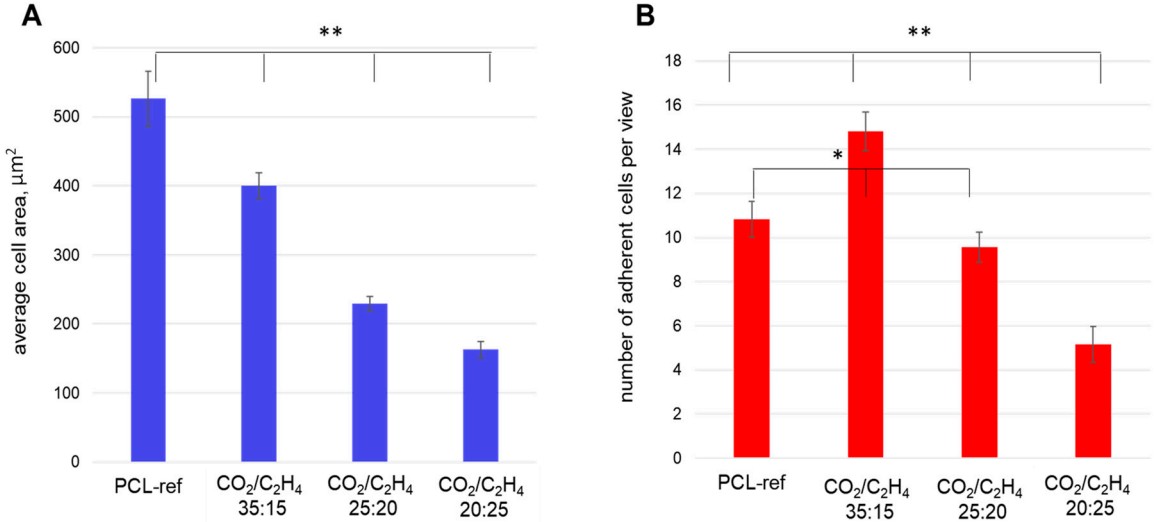

**Figure 9.** The average areas of MSCs (**A**) and number of adherent cells (**B**) measured on different samples: PCL-ref (untreated nanofibers), nanofibers coated by $Ar/CO_2/C_2H_4$ plasma polymers at a $CO_2$:$C_2H_4$ ratios of 35:15, 25:20, and 20:25. The adhesion of cells was assessed by cell area by phalloidin staining of cell actin filaments after 20 min (short-time adhesion). Hoechst staining of the cell nucleus determined the number of cells. ** $p \leq 0.1$, * $p \leq 0.5$.

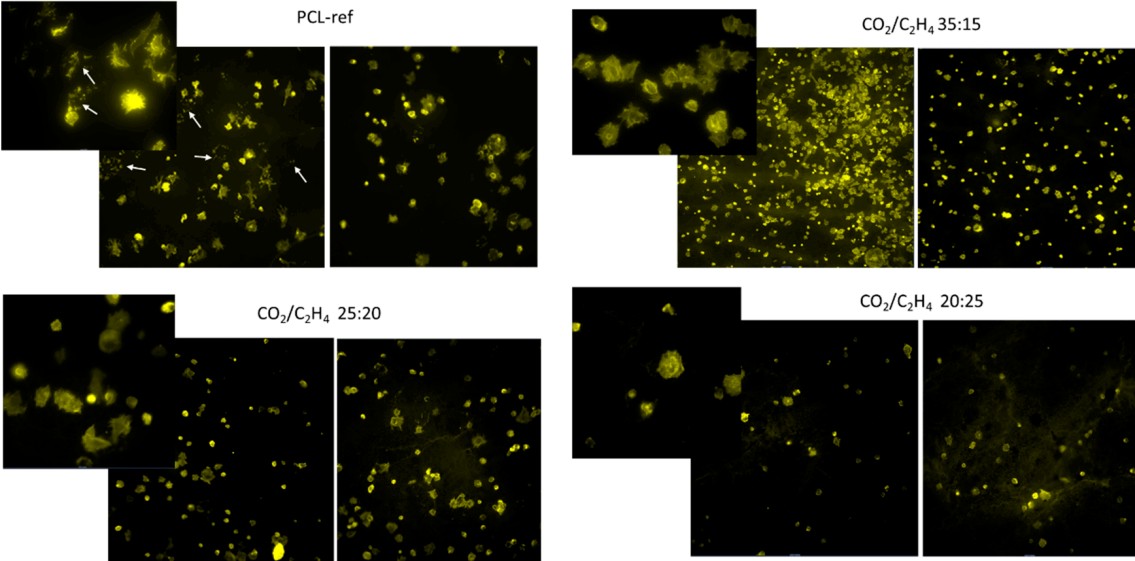

**Figure 10.** Representative fluorescent photographs of cells seeded on nanofibers after 2 h. Actin filaments of the cytoskeleton were stained by Phalloidin. White arrows indicate apoptotic cells. Images are shown at $\times 20$ and $\times 60$ magnification.

Further, the number of cells and their proliferative activity were evaluated after culturing them on the studied surfaces for 72 h. It was shown that the initial defective contact of the seeded cells with the PCL-ref surface affected their subsequent survival and proliferation. After 72 h, cells survived only in cell islands, apparently due to the secretion of growth factors and adhesion molecules by the cells themselves (Figure 11). In turn, cells seeded on the PCL nanofibers coated by plasma layers at CO2:C2H4 ratios of 35:15 and 25:20 were characterized by a smaller spreading area; there are actin rich areas detected upon contact of the processes of the cellular cytoplasm with the surface. This may indicate the formation of good adhesive contacts, andthey were evenly distributed over the entire sample surface (Figure 10). Samples with a high CO2:C2H4 ratio showed a network of well-defined actin microfilaments. Part of the cells had an elongated shape and stress fibrils of most cells began to spread after only 2 h (Figure 10).

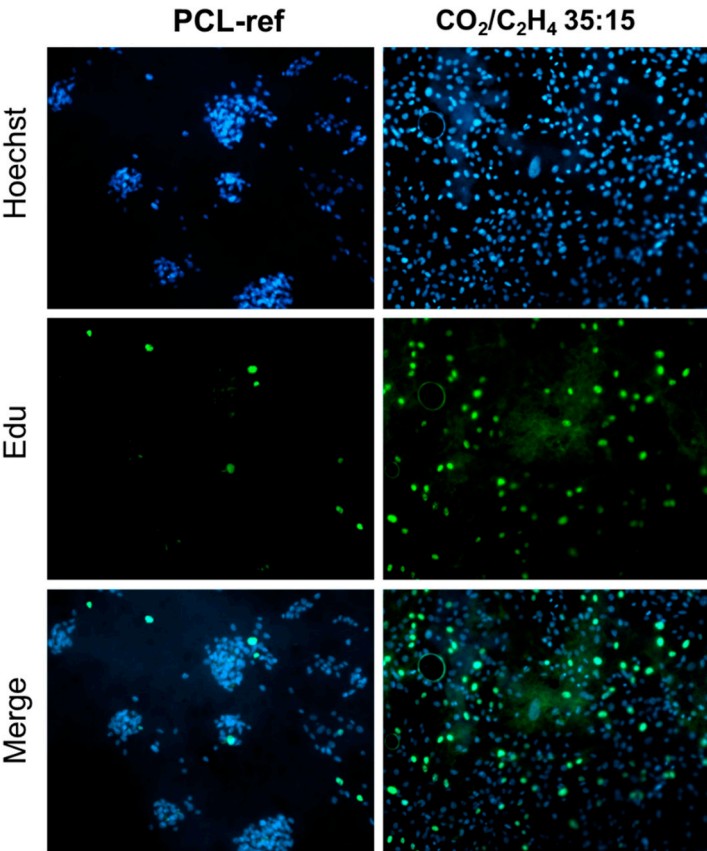

**Figure 11.** Proliferation and viability of human MSCs on the surface of PCL-ref and nanofibers coated with $Ar/CO_2/C_2H_4$ plasma polymer at a $CO_2/C_2H_4 = 35:15$. The cell nucleus is stained with DNA binding fluorescent dye Hoechst 33342 (blue), the proliferating cells are stained with EdUAlexa Fluor™ 488 (green). All images are shown at ×20 magnification.

Nanofibers obtained at a low $CO_2:C_2H_4$ ratio of 20:25 exhibited poor cell adhesion and low cell survival. There were significantly fewer cells on the surface and they had a small spreading area and actin filament networks, while stress fibrils were almost not observed (Figure 10). The estimated values of the cell area are summarized in Figure 9a.

Analysis of the number of adherent cells showed that the sample obtained at $CO_2:C_2H_4 = 35:15$ contained the largest number of live, well-adhered cells (Figure 9b). It was noted that the number of cells on the PCL-ref 2 h after seeding was comparable to samples $CO_2:C_2H_4$ ratio of 20:25, however, Hoechst staining of the nuclei showed hyper condensation of chromatin, a decrease in size and a violation of the shape of nuclei, which indicates apoptotic cell death. This fact is confirmed by the low total number of cells on the PCL-ref sample after 3 days of cultivation (Figure 12).

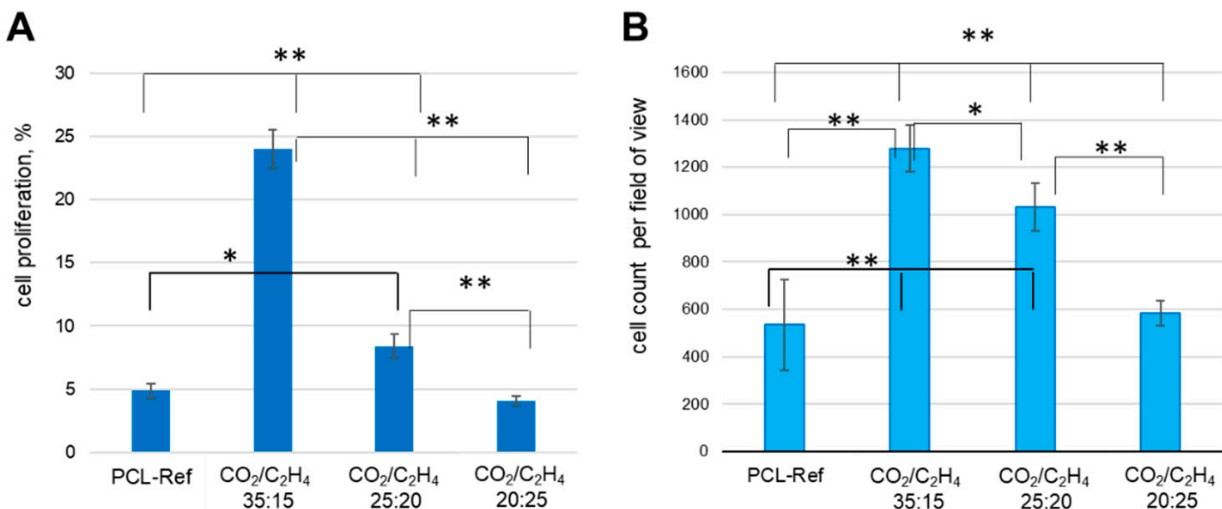

**Figure 12.** (**A**) The percentage of proliferating cells (calculated as the ratio of EdU-positive cells to the total number of Hoechst-positive) and (**B**) Cell count on different samples: PCL-ref (untreated nanofibers), nanofibers coated by $Ar/CO_2/C_2H_4$ plasma polymers at a $CO_2:C_2H_4$ ratios of 35:15, 25:20, and 20:25. ** $p \leq 0.1$, * $p \leq 0.5$.

The maximum percentage of proliferating cells was recorded at a $CO_2:C_2H_4$ ratio of 35:15 compared to 25:20 and 20:25 ($24.1 \pm 1.5$, $8.4 \pm 0.9$, and $4.1 \pm 0.4\%$, respectively). Interestingly, no differences were found between the number of cells on the untreated surface and with the polymer layer obtained at a $CO_2:C_2H_4$ ratios of 20:25 ($4.9 \pm 0.6$ and $4.1 \pm 0.4$, respectively).

## 4. Discussion

Studies of cellular activity and response to various surface modifications of materials are a necessary element of fundamental research in the development of biomedical products. Biocompatibility refers to the ability of nanomaterials (scaffolds) to maintain appropriate cellular activities, including stimulation of molecular and mechanical signaling systems. Surface properties affect morphology, adhesion, proliferation, migration, differentiation, gene expression, and cell metabolism [41]. In general, cells can sense surface topography and align with filopodia. Cytoskeletal actin bundles govern filopodia, which become stable when they encounter favorable surface features. The cell adhesion depends not on only on the surface topography, but more importantly on the surface chemistry, as was shown in numerous research publications [9,45–48]. However, here, we have shown how tremendous changes in cell adhesion and proliferation can be modulated simply by adjusting the concentration of the same surface groups.

If we summarize all our data on the surface properties of $Ar/CO_2/C_2H_4$ plasma polymer layers ($I_{C=O}/I_{CH}$ intensity ratios or concentration of C(O)O contribution), we can see a very clear dependence of the influence of chemical group concentrations on cell adhesion and proliferation. Figure 13 summarizes the number of cells and the percentage of proliferative cells, and we clearly see the positive effect of COOH-rich layers ($Ar/CO_2/C_2H_4$ plasma polymers) on both parameters. A slight decrease for layers with a low COOH densities may be due to the transition from hydrophobic to hydrophilic surface, which leads to loss of protein adhesion due to loss of hydrophobicity. At the same time, poor concentration of COOH groups is not enough to compensate for this effect thanks to COOH (layer)-$NH_2$ (protein) interactions.

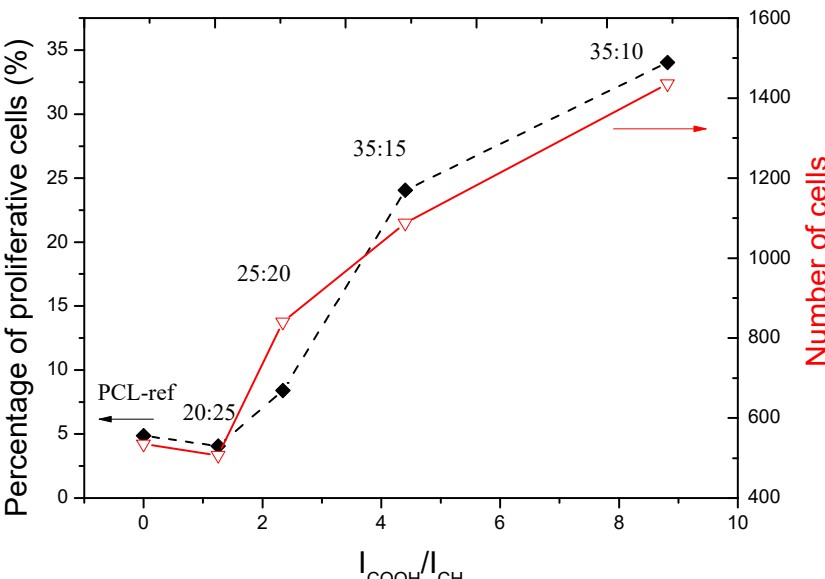

**Figure 13.** The plot of cell numbers and percentage of proliferating cells as a function of $I_{C=O}/I_{CH}$ ratio.

Thus, we have demonstrated that a larger number of cells adhere to the hydrophobic PCL-ref surface and their area is much larger than on more hydrophilic surfaces. After initial adhesion, lamellopodia are formed, moving the cells to the right place. Actin then accumulates within the cell, and filopodial tips form from the nascent focal adhesion points. On surfaces with low adhesive activity, cells cannot find a suitable place for adhesion. This effect was observed earlier [49], and this is a feature of MSCs. It has been shown that for cell proliferation on a hydrophobic surface, intercellular interactions are necessary, while on a superhydrophilic surface, cells adhere and proliferate immediately after seeding [50]. In our earlier studies, we showed that no such effect was found on fibroblasts [51].

Apparently, since MSCs are the precursors of many cell types, their ability to migrate and survive under various conditions differs from differentiated cells (fibroblasts). It has been noted that cells on such hydrophobic surfaces survive exclusively in islets, where they apparently synthesize the components of the extracellular matrix themselves. However, on more hydrophilic surfaces, cells grow evenly. It should be noted that surface signals can cause cells to take on forms corresponding to certain cytoskeletal organizations, which may play an intracellularly role in transduction or enhance some signaling pathways that others do not. These differences may determine the fate of stem cells. Recently, some work has focused on controlling the fate of stem cells through materials design. Therefore, in the future, it is possible to study the effect of various plasma treatment modes on the differentiation potential of MSCs. The difference in the nature of adhesion of different types of cells is also confirmed by other authors [52].

It is generally accepted that cellular responses to a biomaterial are mediated not only by direct contact, but also by an interfacial layer formed on the material surface upon contact with the physiological environment. This interfacial layer is the result of competitive adsorption of proteins from the milieu onto the material surface and depends on the adsorption of proteins as a first step. The protein layer determines the type and extent of subsequent responses [45]. Protein binding may be permanent or temporary, and under certain conditions proteins may be reversibly or irreversibly denatured due to the formation of multiple surface bonds. This situation leads to the breaking of internal bonds within proteins, which leads to denaturation [53]. Perhaps this can explain the fact that, at the maximum number of COOH groups in our study, poor cell adhesion and survival were observed.

## 5. Conclusions

Ensuring good cell adhesion and proliferation to the surface of biodegradable polymers is a challenge. Addressing this important problem, we analyzed the scientific literature and chose the most effective surface treatment method, polymer, and cell types. The most important obtained results can be summarized as follows.

Plasma polymerization in an $Ar/CO_2/C_2H_4$ atmosphere has been successfully applied to synthesize active surface layers with different densities of COOH groups by controlling the $CO_2$:$C_2H_4$ ratio. This method made it possible to deposit a polymer layer with different percentages of C(O)O groups from 5.1 to 14.4%. The water contact angle of polymer layers deposited on PCL nanofibers varies from ~100 to 9°. The adhesion and proliferation of MSCs seeded onto plasma-modified PCL nanofibers was controlled by the composition and wettability of the deposited plasma-polymerized layers. In a sample prepared at a high $CO_2$:$C_2H_4$ ratio, MSCs show a network of well-defined actin microfilaments. Some of the cells have an elongated shape, and the stress fibrils of most cells begin to spread and form actin rings after 2 h. Nanofibers with a polymer layer obtained at a low $CO_2$:$C_2H_4$ ratio (20:25) demonstrate poor cell adhesion and very poor survival. There are much fewer cells on the surface; they have a small spreading area and an undeveloped network of actin filaments, while stress fibrils are almost not observed. The maximum percentage of proliferating cells was recorded for a sample obtained at a $CO_2$:$C_2H_4$ ratio of 35:15 compared to other environments, 25:20 and 20:25 ($24.1 \pm 1.5$, $8.4 \pm 0.9$, and $4.1 \pm 0.4\%$, respectively). Interestingly, no differences are observed between the number of MSCs on the untreated surface and the polymer film deposited at a $CO_2$:$C_2H_4$ ratio of 20:25 ($4.9 \pm 0.6$ and $4.1 \pm 0.4$, respectively). Thus, plasma polymerization in an $Ar/CO_2/C_2H_4$ atmosphere can be considered as an excellent tool for regulating the viability of MSCs by simply adjusting the $CO_2$:$C_2H_4$ ratio.

**Author Contributions:** Conceptualization and methodology, E.S.P., A.M.M., D.V.S. and A.O.S.; biological testing, N.A.S., L.S.K. and A.O.S.; SEM experiments, A.S.K.; XPS, A.M.M.; FT-IR and structure analysis, E.S.P.; plasma experiments, P.V.K.-K.; draft preparation, A.M.M. and E.S.P.; writing—review and editing, D.V.S. All authors have read and agreed to the published version of the manuscript.

**Funding:** This work was supported by the Russian Science Foundation (grant no. 18-75-10057). Part of this work (SEM analyzes) was carried out during the implementation of the strategic project, "Biomedical materials and bioengineering", within the framework of the Strategic Academic Leadership Program "Priority 2030" at NUST MISiS.

**Institutional Review Board Statement:** The study was approved by the Ethics Committee of the RICEL-branch of ICG SB RAS (No 115 from 24 December 2015).

**Informed Consent Statement:** Not applicable.

**Data Availability Statement:** Data is available from corresponding author upon a reasonable request.

**Acknowledgments:** The work was performed using the equipment of the Center for Collective Use «Proteomic Analysis» FRC FTM, supported by funding from the Ministry of Science and Higher Education of the Russian Federation (agreement No. 075-15-2021-691).

**Conflicts of Interest:** The authors declare no conflict of interest.

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
