# Peer review of "Adhesion and Proliferation of Mesenchymal Stem Cells on Plasma-Coated Biodegradable Nanofibers"

_jcs, doi:10.3390/jcs6070193_

Round 1

Reviewer 1 Report

The manuscript entitled “ Adhesion and Proliferation of Mesenchymal Stem Cells on Plasma-coated Biodegradable Nanofibers” by Anton M. Manakhov et al. presents a preparation and characterization of active polymeric surface layers with different densities of COOH groups prepared by Ar/CO2/C2H2 plasma polymerization and the reaction of MSCs on them. The first part of the paper presents the preparation and different characterization of the prepared materials; however, it is hard to follow it because the samples are not clearly named. In one figure, they are called 20:25, 35:15,… and in other some ratio is used and not specified to which sample it belongs. It is easy to count it but for reader quite inconvenient. Other problem is that in most of the text and specifically in biological part, 4 different samples occur – PCL-ref and 3 others, but in some figures (e.g. 4 and 13) 6 or 5 points are presented in the graphs and it is not clear to which sample they are related. As I wrote it is hard to follow characteristics of the materials and cell reaction on them when it is not clearly stated what is what and where. What are the characteristics and cell reaction on PCL-ref is not clear either and thus hard to compare with other samples. This is a major problem of this paper that it is not clear what it presents.

Other major objections

1)      In chapter 3.1 is presented picture 3 and then some numbers related to the nanofiber diameters appear. However, it is not clear, how these numbers were obtained, by which method, how many samples were counted, etc.

2)      Row 249 – it is written that layers did not undergo significant changes in roughness or porosity – how the authors know it, how they measured it and compared it? It is not apparent from the images in fig. 3. According to this figure one may assume that there are pores in a) among fibers but no pores in d). So, this statement can be presented only when there is some support from specific analysis.

3)      Fig. 4 – as I wrote above – there are 6 points in the graph but it is not clear which samples are related to these points (and what is the control PCL-ref).

4)      In “chemical” part of the paper is sample 35:15 but in “biological” part 30:15. Is it the same sample or two different? If so, then why two different samples were used?

The biological part does not present in the figures what is written in the text. The text and figures are not logically ordered and it is hard to follow the presented “story”.

5)      In the Methods part, it is written that cells were seeded for 20 min and stained for phalloidin, but in the Results part is figure 9 where it is written that phalloidin was detected after 2 h. The text in 3.3 Chapter is not logically structured, it starts with size of cells but this is presented later then the picture of the cells. Moreover, the images in figure 9 are not well presented, the scale bar is missing and the magnification is not appropriate for showing that some cells are bigger than others are. From presented pictures is clear that there are a lot of cells in sample 35:15 but it is all what one can say by naked eye from the picture. The arrow indicating “apoptotic” cells can point to apoptotic cell but also it can be missleading. By phalloidin staining one cannot clearly identify apoptosis. For that other methods should be used and then can be stated that apoptosis occurs.

6)      Row 344 – it is written that the cells formed good adhesive contacts – how they know it, for this statement staining of focal adhesion must be done and image analysis of the obtained images, however, it is not presented elsewhere.

7)      Actin ring is not specific hallmark for well adhered MSCs, it is a hallmark for osteoclast.

8)      In row 355-356 is written – Although the maximum number of cells was on PCL-ref, cells were MORPHOLOGICALLY DEAD – What does this mean? Where is some experiment and method proving this statement?

9)      Fig. 10 – the time point of this analysis is missing. It will be better to present separately proliferating cells (green staining) and all cells (blue staining) to see by naked eye if the cell really proliferate. On the other hand, used Hoechst 33342 is specific indicator of apoptosis but results (graph) showing the outcome of this staining is not presented elsewhere.

10)  Fig. 11 and 12 compare the cell reaction to different samples but because of missing statistics, it is hard to comment these results.

11)  Fig. 12A – it presents cell proliferation in % but it is not clear what is 100%. It shows that the best proliferation was obtained on sample 30:15 but 25% of proliferating cells is really not big deal. According to this result, the cells do not like the surface and thus it is not good for further usage.

12)  Fig. 13 – it is not clear which dot represents PCL-ref and other samples and from y-axis on the right side it is not clear which number of cells (on the view, on cm2,…?)

Discussion part

13)  This part cannot be called discussion; it does not discussed anything and is not supported by other papers who work on similar topics. It is just description of results in more literary form. Moreover, many speculations are presented there but not supported by presented results or some literature findings. In addition, it comments mostly the biological part but nothing about the chemical part of the paper.

Minor objections

The text is not polished from abundant (not deleted) characters and sentences (row 215-229).

Author Response

We are grateful to Reviewr for his/her great comments that definitely were very essential to improve the quality of our paper. The response to the comments are provided in the PDF file, while all changes were tracked in the revised submission.

Reviewer 1

Comment 1: The first part of the paper presents the preparation and different characterization of the prepared materials; however, it is hard to follow it because the samples are not clearly named. In one figure, they are called 20:25, 35:15,… and in other some ratio is used and not specified to which sample it belongs. It is easy to count it but for reader quite inconvenient.

Author response

The sample designations were added on Figure 4, 6, 13

Comment 2: Other problem is that in most of the text and specifically in biological part, 4 different samples occur – PCL-ref and 3 others, but in some figures (e.g. 4 and 13) 6 or 5 points are presented in the graphs and it is not clear to which sample they are related. As I wrote it is hard to follow characteristics of the materials and cell reaction on them when it is not clearly stated what is what and where. What are the characteristics and cell reaction on PCL-ref is not clear either and thus hard to compare with other samples. This is a major problem of this paper that it is not clear what it presents.

 Author response

How it was described in Materials and Methods, changing CO2:C2H4 ratio 5 types of plasma-deposited polymers were obtained

“The samples with CO2:C2H4 ratios of 40:5, 35:10, 35:15, 25:20 and 20:25 were prepared and investigated. They are denoted according to the CO2:C2H4 flows ratio.”

On figure 4 and 13 the additional 6 point corresponds to untreated PCL NFs (PCL-ref)

The sample designations were added on Figure 4, 6, 13

          Comment 3: In chapter 3.1 is presented picture 3 and then some numbers related to the nanofiber diameters appear. However, it is not clear, how these numbers were obtained, by which method, how many samples were counted, etc.

       Author response

The next sentence was added in chapter 2.3: “The average NFs diameter was determined using the ImageJ software based on 100 measurements.”

       Comment 4: Row 249 – it is written that layers did not undergo significant changes in roughness or porosity – how the authors know it, how they measured it and compared it? It is not apparent from the images in fig. 3. According to this figure one may assume that there are pores in a) among fibers but no pores in d). So, this statement can be presented only when there is some support from specific analysis.

            Author response

The phrase:Since the morphology of the Ar/CO2/C2H4 plasma polymerized layers did not undergo significant changes in roughness or porosity” was deleted

      Comment 5: Fig. 4 – as I wrote above – there are 6 points in the graph but it is not clear which samples are related to these points (and what is the control PCL-ref).

             Author response

      Changing CO2:C2H4 ratio 5 types of plasma-deposited polymers were obtained

     “The samples with CO2:C2H4 ratios of 40:5, 35:10, 35:15, 25:20 and 20:25 were prepared and investigated. They are denoted according to the CO2:C2H4 flows ratio.”

      On figure 4 and 13 the additional 6 point corresponds to untreated PCL NFs (PCL-ref) and corresponds zero point, since no treatment (no flow CO2 or C2H4) was applied

       Comment 6: In “chemical” part of the paper is sample 35:15 but in “biological” part 30:15. Is it the same sample or two different? If so, then why two different samples were used?

 Author response

It was the same sample, the signature was corrected.

Comment 7: The biological part does not present in the figures what is written in the text. The text and figures are not logically ordered and it is hard to follow the presented “story”.

Author response

Сorresponding changes have been made to the text

        Comment 8: In the Methods part, it is written that cells were seeded for 20 min and stained for phalloidin, but in the Results part is figure 9 where it is written that phalloidin was detected after 2 h.

Thanks for the note, corrected

The text in 3.3 Chapter is not logically structured, it starts with size of cells but this is presented later then the picture of the cells. Moreover, the images in figure 9 are not well presented, the scale bar is missing and the magnification is not appropriate for showing that some cells are bigger than others are. From presented pictures is clear that there are a lot of cells in sample 35:15 but it is all what one can say by naked eye from the picture. The arrow indicating “apoptotic” cells can point to apoptotic cell but also it can be missleading. By phalloidin staining one cannot clearly identify apoptosis. For that other methods should be used and then can be stated that apoptosis occurs.

 Author response

We agree with the reviewer that phalloidin staining does not detect apoptosis. We detect apoptosis by staining nuclei with Hoechst. Figure 11 is present to demonstrate the shape of cells without stained nuclei. In this case, the arrows point to membrane`s fragments of poorly adhered cell that have peeled off the surface. The text has been corrected accordingly.

        Comment 9: Row 344 – it is written that the cells formed good adhesive contacts – how they know it, for this statement staining of focal adhesion must be done and image analysis of the obtained images, however, it is not presented elsewhere.

Author response

Text has been corrected: there are detected actin rich areas upon contact of the processes of the cellular cytoplasm with the surface. This may indicate the formation of good adhesive contacts, and were evenly distributed over the entire sample surface

      Comment 10: Actin ring is not specific hallmark for well adhered MSCs, it is a hallmark for osteoclast.

Author response

      Comment 11: In row 355-356 is written – Although the maximum number of cells was on PCL-ref, cells were MORPHOLOGICALLY DEAD – What does this mean? Where is some experiment and method proving this statement?

Author response

We determined the dead cells on the fibers by the Hoechst staining of the nuclei, we agree that the wording «MORPHOLOGICALLY DEAD» is not correctly applied. The text has been changed:. It was noted, that the number of cells on the PCL-ref 2 hours after seeding was comparable to samples СO2:C2H4 ratio of 20:25, however, Hoechst staining of the nuclei showed hyper condensation of chromatin, a decrease in size and a violation of the shape of nuclei, which indicates apoptotic cell death. This fact is confirmed by the low total number of cells on the PCL-ref sample after 3 days of cultivation     

Comment 12: Fig. 10 – the time point of this analysis is missing. It will be better to present separately proliferating cells (green staining) and all cells (blue staining) to see by naked eye if the cell really proliferate. On the other hand, used Hoechst 33342 is specific indicator of apoptosis but results (graph) showing the outcome of this staining is not presented elsewhere.

Author response

Fluorescent photographs are divided into channels and added to the article. Indeed, we use Hoechst staining to identify apoptotic cells. In order not to overload the article, we used only the parameters of the total number of cells determined after 72 hours on the studied materials and their proliferative activity.

      Comment 13: Fig. 11 and 12 compare the cell reaction to different samples but because of missing statistics, it is hard to comment these results.

Author response

Statistics added

      Comment 14: Fig. 12A – it presents cell proliferation in % but it is not clear what is 100%. It shows that the best proliferation was obtained on sample 30:15 but 25% of proliferating cells is really not big deal. According to this result, the cells do not like the surface and thus it is not good for further usage.

Author response

The % of proliferation of cells calculated as the ratio of EdU-positive cells to the total number of Hoechst-positive. A fairly low percentage of proliferation (25%) is indeed not high for MSCs, however, as can be seen in the photo (Fig), the cell density is quite high, so contact inhibition of proliferation occurs.

      Comment 15: Fig. 13 – it is not clear which dot represents PCL-ref and other samples and from y-axis on the right side it is not clear which number of cells (on the view, on cm2,…?)

 Author response

Corrected

Comment 16 (Discussion part): This part cannot be called discussion; it does not discussed anything and is not supported by other papers who work on similar topics. It is just description of results in more literary form. Moreover, many speculations are presented there but not supported by presented results or some literature findings. In addition, it comments mostly the biological part but nothing about the chemical part of the paper.

Author response

The next paragraph was added to the discussion chapter

Comment 17: The text is not polished from abundant (not deleted) characters and sentences (row 215-229).

Author response

The abundant sentences (row 215-229) were deleted.

Reviewer 2 Report

The authors present a study entitled "Adhesion and proliferation of mesenchymal stem cells on plasma-coated biodegradable nanofibers." In this study, plasma polymerization in Ar/CO2/C2H4 environment has been successfully applied to synthesize active surface layers with different densities of COOH groups by controlling the CO2:C2H4 ratio. The results are well analyzed and conclusive, but some comments should be addressed.

·       The authors should explain in more details how nanofibers play an important role in cell adhesion and proliferation.

·       On what parameters was PCL selected as the electrospinning polymer in this study?

·       Since this study focused on the surface properties of nanofibers, it would be preferable that the authors conduct cell studies with more than one cell line.

·       The manuscript contained additional information from the journal instruction that is not related to this study. Please check line numbers 215 to 229 on page 6. It should be removed.

·       Since this research article is not a review article, are Figures 1 and 2 necessary? If necessary, how about moving to SI?

Author Response

We are are grateful to reviewer for his/her comments. Our answers are below in point-by point style.

Comment 1: The authors should explain in more details how nanofibers play an important role in cell adhesion and proliferation.

Author response

A tremendous amount of recent reviews [1–4]demonstrated the high potential of using biodegradable nanofibers for tissue engineering (TE) including bone TE[4,5], vascular TE[6], skin TE[7], meniscus [8] and etc. Electrospun nanofibers possess characteristics such as porosity, conformity, and interconnected architecture, mechanical properties. The biocompatibility and biodegradability properties depend on the polymer, which was used to form nanofibers. Compared to other available biodegradable polymers, PCL could offer distinct advantages, thereby surpassing other candidates for biomedical applications. For instance, customizable degradation rate and mechanical properties, simple shaping and production that could allow for creation of pore sizes suitable for tissue ingrowth, and the possibility of drug delivery in a controlled manner are important merits of PCL based systems.[9] Moreover, nanofibers mimicking the architecture of the natural extracellular matrix (ECM) what influence on cell proliferation, adhesion,toxicity, and growth behavior while minimizing a negative effect on cell growth.

Huang et al. [10] compared of 3T3 mouse fibroblast cell behavior on pva-gelatin electrospun nanofibers with random and aligned configuration. They demonstrated that configuration influence on cell growing:  random configuration of fibers lead to stochastically distributed of cells, while aligned configuration of fibers lead to cells growing on assembled in alignment with the direction of the nanofibers, resulting in a uniform linear pattern coinciding with the polarized cell growth. However, the stretching of the cell is coupled with nuclear deformation caused by the lateral pulling stress from the F-actin fiber transmitted to the cell nucleus, thus, random configuration of NFs is preferable.

  1. Ehrmann, A. Non-Toxic Crosslinking of Electrospun Gelatin Nanofibers for Tissue Engineering and Biomedicine — A Review. Polymers (Basel). 2021, 13, 1973.
  2. Nirwan, V.P.; Kowalczyk, T.; Bar, J.; Buzgo, M.; Filov, E. Advances in Electrospun Hybrid Nanofibers for Biomedical Applications. Nanomaterials 2022, 12, 1829.
  3. Stocco, T.D.; Jos, P.; Augusto, M.; Filho, D.A.; Lobo, A.O. Nanohydroxyapatite Electrodeposition onto Electrospun Nanofibers : Technique Overview and Tissue Engineering Applications. Bioengineering 2021, 8, 151.
  4. Smith, J.A.; Mele, E. Electrospinning and Additive Manufacturing : Adding Three-Dimensionality to Electrospun Scaffolds for Tissue Engineering. Front. Bioeng. Biotechnol. 2021, 9, 1–7, doi:10.3389/fbioe.2021.674738.
  5. Raja, I.S.; Preeth, D.R.; Vedhanayagam, M.; Hyon, S.; Lim, D. Polyphenols-loaded electrospun nanofibers in bone tissue engineering and regeneration. Biomater. Res. 2021, 35, 1–16.
  6. Rickel, A.P.; Deng, X.; Engebretson, D.; Hong, Z. Electrospun nanofiber scaffold for vascular tissue engineering. Mater Sci Eng C Mater Biol Appl. 2021, 129, 112373, doi:10.1016/j.msec.2021.112373.Electrospun.
  7. Yu, R.; Zhang, H.; Guo, B. Conductive Biomaterials as Bioactive Wound Dressing for Wound Healing and Skin Tissue Engineering. NanoMicro Lett. 2022, 14, 1–46, doi:10.1007/s40820-021-00751-y.
  8. Wang, X.; Ding, Y. Advances in electrospun scaffolds for meniscus tissue engineering and regeneration. J Biomed Mater Res 2022, 110, 923–949, doi:10.1002/jbm.b.34952.
  9. Homaeigohar, S.; Boccaccini, A.R. Nature-Derived and Synthetic Additives to poly ( ε -Caprolactone ) Nano fi brous Systems for Biomedicine ; an Updated Overview. Front. Chem. 2022, 9, 1–25, doi:10.3389/fchem.2021.809676.
  10. Huang, C.; Hu, K.; Wei, Z. Comparison of cell behavior on pva / pva-gelatin electrospun nanofibers with random and aligned configuration. Sci. Rep. 2016, 6, 37960–37968, doi:10.1038/srep37960.

Comment 2: On what parameters was PCL selected as the electrospinning polymer in this study?

Author response

Compared to other available biodegradable polymers, PCL could offer distinct advantages, thereby surpassing other candidates for biomedical applications. For instance, customizable degradation rate and mechanical properties, simple shaping and production that could allow for creation of pore sizes suitable for tissue ingrowth, and the possibility of drug delivery in a controlled manner are important merits of PCL based systems.[9]

Homaeigohar, S.; Boccaccini, A.R. Nature-Derived and Synthetic Additives to poly ( ε -Caprolactone ) Nano fi brous Systems for Biomedicine ; an Updated Overview. Front. Chem. 2022, 9, 1–25, doi:10.3389/fchem.2021.809676.

Comment 3: Since this study focused on the surface properties of nanofibers, it would be preferable that the authors conduct cell studies with more than one cell line.

Author response

This work is devoted to the analysis of various modes of modifications of polycaprolactone nanomaterials. Screening and selection of the most suitable for biomedical applications was carried out using MSCs. In the future, they will be analyzed in more detail on various cell lines using a wider range of methods.

Comment 4: The manuscript contained additional information from the journal instruction that is not related to this study. Please check line numbers 215 to 229 on page 6. It should be removed.

Author response

The abundant sentences (row 215-229) were deleted. 

Comment 5:Since this research article is not a review article, are Figures 1 and 2 necessary? If necessary, how about moving to SI?

We do prefer to keep these Figs in the main part because it nicely shows the way how we selected the materials and processes. And it is also good to show the “Big Picture”

Reviewer 3 Report

It is an interesting and high-quality paper that suggests a way to control the degree of adhesion and proliferation of stem cells through surface treatment of PCL. However, there are some explanations or data that need to be added for the publication of the paper.

 1. Please include a specific explanation of why you chose PCL and its application

 2. It would be great if you could add data on the changes in the mechanical properties and decomposition behavior of PCL before and after plasma treatment.

 3. It is known that -COOH group is also produced by surface treatment by PCL hydrolysis. Please explain the differences and merits and demerits

 4. The increase in cell proliferation and adhesion seems to require more specific data and explanation as to whether it is due to an increase in -COOH group or changes in surface roughness and polarity of the surface.

Author Response

We want to thank the reviewer for his/her great comments that very helped us.

Please find our answers below.

  1. Please include a specific explanation of why you chose PCL and its application

Answer

Thank you for this comment. We added a sentence on Page 2

Indeed, PCL possesses good mechanical properties and long-term stability from a few months up to 3 years in vivo and it is approved by U.S. Food and Drug Administration. Thus, recently PCL is becoming the material of choice for biomedical materials.

  1. It would be great if you could add data on the changes in the mechanical properties and decomposition behavior of PCL before and after plasma treatment.

Answer

The mechanical properties of PCL nanofibers was studied before and the plasma modified PCL can have even better mechanical properties. Please se the ref40 https://www.mdpi.com/2073-4360/12/6/1403/htm

However, in this work we did not provided the mechanical testing. We will do it in our future reseach.

  1. It is known that -COOH group is also produced by surface treatment by PCL hydrolysis. Please explain the differences and merits and demerits

Answer

Indeed, the chemical treatment of PCL may induce the hydrolysis of ester bonds and lead to the formation of COOH groups. Nevertheless, this wet chemistry approach is not environment-friendly (as we explained in the Introduction) and, moreover, the wet chemistry would affect the bulk properties of PCL.

  1. The increase in cell proliferation and adhesion seems to require more specific data and explanation as to whether it is due to an increase in -COOH group or changes in surface roughness and polarity of the surface.

Answer

We have performed the SEM analysis and the micrographs of the PCL nanofibers modified with various plasma polymer layers (with different CO2/C2H4 ratio) exhibited very similar structures.

Round 2

Reviewer 1 Report

The authors have significantly improved the Results part (biological) of the paper with clear naming of samples in the graphs and text, statistics addition and comments related to what they clearly present and not what they feel.

Despite this I have some more comments. Cell number and area was according to Methods part analysed by IN Cell Analyzer 2200 at 600 magnification. I do not understand what this magnification means. I think that it is good for seeing clearly the cell size (fig.9 A) but not for cell counting. Having 10 cells on image  and less (fig.9B) in not appropriate for counting the cell numbers and compare them with each other. From which magnification and “analyser” comes images from fig. 10?  There are for sure more cells per image than presented 5-15 in Fig. 9B. I suggest to count cell number from the rightmost picture of each sample in fig 10 and then I can believe that obtained cell numbers are valid and significant. According to these pictures from fig. 10 I have the feeling that the most cells are on 35:15 followed by 25:20 and followed by PCL-ref, but it is different outcome in comparison to the fig. 9B graph.

Another comment the way of adhesion of cells to very hydrophobic PCL-ref and very hydrophilic 35:15 can be totally different and mediated by different mechanism and needs some discussion. For example in Verdanova et al, 2017 (doi 10.1007/s00418-017-1571-7) is shown that cell area of MSC is comparable on the surface coated with proteins from FBS and without these proteins, however, cell number varies greatly.

Despite the fact, I called for SOME discussion and not only summary of the results, in this Version 2 the discussion is still missing.

Author Response

First of all, we are grateful to reviewer for his valuable comments. We improved our discussion section, add more refs and now it is about two pages of discussion of previous works and comparison with our achievements. Please find our answers below, and all new correction are highlighted in yellow color in our new version.

Comment 1

Despite this I have some more comments. Cell number and area was according to Methods part analysed by IN Cell Analyzer 2200 at 600 magnification. I do not understand what this magnification means. I think that it is good for seeing clearly the cell size (fig.9 A) but not for cell counting. Having 10 cells on image  and less (fig.9B) in not appropriate for counting the cell numbers and compare them with each other. From which magnification and “analyser” comes images from fig. 10?  There are for sure more cells per image than presented 5-15 in Fig. 9B. I suggest to count cell number from the rightmost picture of each sample in fig 10 and then I can believe that obtained cell numbers are valid and significant. 

Author response

We thank the reviewer for the comment. Indeed, the number and average size of cells were assessed at a finer magnification (x20), x60 magnification was used to better visualize the shape of the cells, the formation of actin fibers. Necessary changes have been made to the main text (section Materials and Methods, as well as magnification captions have been added to Fig. 10).

According to these pictures from fig. 10 I have the feeling that the most cells are on 35:15 followed by 25:20 and followed by PCL-ref, but it is different outcome in comparison to the fig. 9B graph.

Author response

The graphs show the statistically processed results of calculating 36 fields of view on each substrate, presented in three repeats

Reviewer 2 Report

Accept in present form

Author Response

We are grateful to the reviewer for his great input and his high mark for our work.

Reviewer 3 Report

I think it can be published in a journal

Author Response

We are very thankful to the reviewer for his great input and high mark for our work.

Round 3

Reviewer 1 Report

The paper in this current form includes all the information necessary for interesting paper.